# 7S,15R-Dihydroxy-16S,17S-Epoxy-Docosapentaenoic Acid, a Novel DHA Epoxy Derivative, Inhibits Colorectal Cancer Stemness through Repolarization of Tumor-Associated Macrophage Functions and the ROS/STAT3 Signaling Pathway

**DOI:** 10.3390/antiox10091459

**Published:** 2021-09-14

**Authors:** Lifang Wang, Hack Sun Choi, Yan Su, Binna Lee, Jae Jun Song, Yong-Suk Jang, Jeong-Woo Seo

**Affiliations:** 1Microbial Biotechnology Research Center, Korea Research Institute of Bioscience and Biotechnology (KRIBB), Jeongeup-si 56212, Korea; zmz0@kribb.re.kr (L.W.); zsr4@kribb.re.kr (Y.S.); skanwodd@kribb.re.kr (B.L.); jjsong@kribb.re.kr (J.J.S.); 2Department of Bioactive Material Sciences and The Institute for Molecular Biology and Genetics, Jeonbuk National University, Jeonju 54896, Korea; 3Faculty of Biotechnology, College of Applied Life Sciences, Jeju National University, Jeju 63243, Korea; choix074@jejunu.ac.kr

**Keywords:** 7,15,16,17-epoxy-tetrahydroxy docosahexaenoic acid (diHEP-DPA), tumor-associated macrophages, epithelial-mesenchymal transition (EMT), colorectal cancer stemness, ROS, STAT3

## Abstract

Colorectal cancer is a highly malignant cancer that is inherently resistant to many chemotherapeutic drugs owing to the complicated tumor-supportive microenvironment (TME). Tumor-associated macrophages (TAM) are known to mediate colorectal cancer metastasis and relapse and are therefore a promising therapeutic target. In the current study, we first confirmed the anti-inflammatory effect of 7S,15R-dihydroxy-16S,17S-epoxy-docosapentaenoic acid (diHEP-DPA), a novel DHA dihydroxy derivative synthesized in our previous work. We found that diHEP-DPA significantly reduced lipopolysaccharide (LPS)-induced inflammatory cytokines secretion of THP1 macrophages, IL-6, and TNF-α. As expected, diHEP-DPA also modulated TAM polarization, as evidenced by decreased gene and protein expression of the TAM markers, CD206, CD163, VEGF, and TGF-β1. During the polarization process, diHEP-DPA treatment decreased the concentration of TGF-β1, IL-1β, IL-6, and TNF-α in culture supernatants via inhibiting the NF-κB pathway. Moreover, diHEP-DPA blocked immunosuppression by reducing the expression of SIRPα in TAMs and CD47 in colorectal cancer cells. Knowing that an inflammatory TME largely serves to support epithelial-mesenchymal transition (EMT) and cancer stemness, we tested whether diHEP-DPA acted through polarization of TAMs to regulate these processes. The intraperitoneally injected diHEP-DPA inhibited tumor growth when administered alone or in combination with 5-fluorouracil (5-FU) chemotherapy in vivo. We further found that diHEP-DPA effectively reversed TAM-conditioned medium (TCCM)-induced EMT and enhanced colorectal cancer stemness, as evidenced by its inhibition of colorectal cancer cell migration, invasion and expression of EMT markers, as well as cancer cell tumorspheres formation, without damaging colorectal cancer cells. DiHEP-DPA reduced the population of aldehyde dehydrogenase (ALDH)-positive cells and expression of colorectal stemness marker proteins (CD133, CD44, and Sox2) by modulating TAM polarization. Additionally, diHEP-DPA directly inhibited cancer stemness by inducing the production of reactive oxygen species (ROS), which, in turn, reduced the phosphorylation of nuclear signal transducer and activator of transcription 3 (STAT3). These data collectively suggest that diHEP-DPA has the potential for development as an anticancer agent against colorectal cancer.

## 1. Introduction

Colorectal cancer is a highly malignant cancer that, with the aging of the population and changes in lifestyle, has come to rank third in global cancer-related deaths [1,2]. In 2020, an estimated 1.9 million new cases were diagnosed and 935,000 associated deaths were recorded; notably, malignant tumor progression and metastases lead to the high mortality of colorectal cancer [3]. Although chemotherapy, targeted therapy, and combinations of these therapies have substantially improved prognosis and prolonged survival, they may also promote tumor growth and migration, trigger a cytokine storm in the tumor microenvironment (TME), as well as activate the macrophage production of proinflammatory mediators, all of them may promote tumor progression [4]. Cancer cells are embedded in the TME, and immune components of the TME can modulate tumor progression. Macrophages play a critical role in the immune cell mediators of chronic inflammation related to tumor formation; it acts as a “bridge” connecting inflammation with malignant cell transformation. Macrophages are classified as either an M1 or M2 subtype depending on their immune responses [5]. M1-polarized macrophages mediate acute inflammation and play a role in inflammation-induced mutagenesis. M2-polarized macrophages inhibit acute inflammation, while can promote tumor aggravation. There are four subtypes in M2-like macrophages: M2a, b, c, and d [6]. In general, M2a, M2b, and M2c macrophages tend to reduce inflammation, whereas M2d macrophages tend to infiltrate the TME and thus are collectively referred to as tumor-associated macrophages (TAMs) [7]. TAMs are among the most highly represented cell populations in the TME and have important roles in invasive, angiogenic and metastatic processes. The activation modes and markers of TAMs in different tumor tissues are diverse; thus, it is still not clear exactly what induces TAMs, but possibilities include combinations of vascular endothelial growth factors (VEGFs), tumor growth factor (TGF)-β, interleukin (IL)-4, C-C motif chemokine ligand 2 (CCL2), colony-stimulating factors (CSFs), and some extracellular matrix components. Despite their heterogeneity, different TAMs play similar tumor-promoting roles in the microenvironment by secreting cytokines (e.g., IL-10, VEGFs, and EGF) [8,9]. Another mechanism of tumor immune evasion is CD47, a critical “don’t eat me” signal for the innate immune system and regulator of the adaptive immune response that is over-expressed on the surface of most tumors [10]. CD47 interacts with signal regulatory protein (SIRP)-α, which enables tumors to avoid innate immune surveillance [11]. Some reports have shown that TAM infiltration is usually positively correlated with SIRPα expression, which might be explained by stimulation of SIRPα expression in macrophages by CD47 self-antigen expressed in tumors, although the extent of this stimulation varies depending on tumor type [12].

In addition to tumor immune evasion, the recently proposed “cancer stem cell” theory posits that cancer stem cells (CSCs) also ultimately result in resistance to treatment and cancer recurrence [13,14]. CSCs are resistant to current therapies compared with cancer cells because most of these cells are maintained in a resting state, whereas existing therapies target proliferating tumor cells [15,16,17]. Colorectal CSCs were found to be inherently resistant to first-line chemotherapeutic agents for colorectal cancer, such as 5-fluorouracil (5-FU) and oxaliplatin, among others. In many colorectal cancer patients, they can initially achieve a good therapeutic effect. However, multidrug resistance soon manifests, followed by rapid tumor progression, which may reflect the inability of chemotherapy agents to kill these non-proliferating CSCs despite killing proliferating tumor cells. Tumor-associated inflammatory factors play an important role in the self-renewal of cancer cell stemness and tumor malignancy [18,19,20,21]. Subsets of colorectal CSCs expressing different molecular markers, including ALDH1 (aldehyde dehydrogenase-1), LGR5 (leucine-rich repeat-containing G protein-coupled receptor 5), as well as the surface antigens CD133, CD44, CD166, CD29, and CD24, have been discovered, and it has been found that clones formed from single cells expressing these molecular markers can all form tumors. Increasingly, studies have begun to focus on the molecular mechanisms that confer these survival advantages on CSCs.

In the epithelial–mesenchymal transition (EMT) process, epithelial cancer cells lose their epithelial characteristic and acquire the features of mesenchymal cells by reducing epithelial markers (e.g., E-cadherin) and increasing mesenchymal markers (Vimentin and N-cadherin). Some reports have concluded that cancer cells that have undergone EMT express high levels of stem surface markers [22]. Recent studies have reported that TAMs are associated with EMT in breast [21], lung [23], and pancreatic cancer [24] by virtue of their secretion of inflammatory cytokines, and enhance cancer stemness through EMT. However, the mechanisms by which TAMs promote EMT and cancer stemness in colorectal cancer cells remains unknown.

Specialized pro-resolving mediators (SPMs) are a superfamily of endogenous chemical mediators, including resolvins, protectins, and maresins [25,26]. An inflammatory TME is a major contributor to cancer cell stemness, metastasis, relapse, and negative outcomes in cancer patients. Thus, the resolution of inflammation is crucial for efforts to reduce the cancer stemness of CSCs. Among resolvins, resolvin Ds have been reported to be beneficial in various cancers, including lung, liver, pancreas, stomach, and colon cancer [27,28,29,30,31,32,33,34,35,36,37]. Resolvin D1 has been reported to inhibit hyper-expressed c-Myc by attenuating its phosphorylation-dependent stabilization in HCT116 colon cancer cells, and has also been shown to prevent the progression of hepatitis to liver cancer [30,38]. Resolvin D1 and Resolvin D2 at low concentrations (100 pM) were shown to inhibit the adherence and proliferation of a human oral squamous cell carcinoma cell line (HSC-3) in vitro [39]. Resolvin D1, Resolvin D2, and Resolvin E1 were found to inhibit debris-stimulated inflammatory prostate, breast, and liver tumors [36]. The antitumor activities and mechanisms of resolvins differ by cancer type, and a previous study suggested that resolvins exert their antitumor activity through stromal cells instead of directly acting on tumor cells. The clearance of debris via macrophage-mediated phagocytosis modulation of macrophage polarization of M1 and M2 type macrophages and TAMs might be enhanced by resolvins [36,39,40]. Resolvin D1 also prevents EMT and reduces the stemness of hepatocellular carcinoma via restraining the paracrine action of cancer-associated fibroblasts, and inhibits EMT in A549 lung cancer cells [31]. Moreover, aspirin-induced Resolvin D1 was shown to decrease EMT through inhibition of the mTOR pathway, which is closely linked to oxidative stress [40,41]. Resolvin D1 and D2 suppressed prostate tumor growth and inflammation by inhibiting the polarization of TAMs [42].

7S,15R-Dihydroxy-16S,17S-epoxy-docosapentaenoic acid (diHEP-DPA) is a novel resolvin synthesized from the substrate, docosahexaenoic acid (DHA) as presented in our previous work and the structure of diHEP-DPA by liquid chromatography–tandem mass spectrometry (LC-MS/MS) and nuclear magnetic resonance (NMR) is also reported earlier [43]. In the current study, we further studied the effects of diHEP-DPA on TAM polarization, immune suppression, EMT, and CSCs in the TME. Our results collectively indicate that diHEP-DPA inhibits colorectal cancer growth in vitro and in vivo, suggesting its potential for development as an anticancer agent that acts on the TME.

## 2. Materials and Methods

### 2.1. Materials

7S,15R-Dihydroxy-16S,17S-epoxy-docosapentaenoic acid (diHEP-DPA) was obtained from DHA through an enzymatic reaction using cyanobacterial lipoxygenase and purified (purity > 98%) as previously described [43]. The human monocytic cell line THP-1, human colorectal cancer cells (HT29 and HCT116), and mouse colorectal cancer cell line CT26 were purchased from Korea Cell Line Bank (KCLB, Seoul, Korea). Phorbol 12-myristate 13-acetate (PMA), lipopolysaccharide (LPS), N-acetylcysteine (NAC), and 5-FU were obtained from Sigma-Aldrich Co. (St. Louis, MO, USA). Cell viability was assayed using a commercial CellTiter 96 AQueous One Solution kit (Promega, Madison, WI, USA). Human inflammatory cytokines were assayed by a BDTM Cytometric Bead Array human inflammatory cytokine assay kit (BD Biosciences, San Jose, CA, USA). Cancer cell apoptosis was assayed using an Annexin V/Propidium Iodide (AV/PI) Kit (BD Biosciences, San Jose, CA, USA). An ALDEFLUOR Kit (Stemcell Technologies Inc., Vancouver, BC, Canada) was used to assay ALDH1 activity. DiHEP-DPA was stored at −20 °C in 100% dimethyl sulfoxide (DMSO).

### 2.2. Determination of the Cytokines IL-6, TNF-α, VEGF and TGF-β1

THP-1 cells were grown in RPMI-1640 medium (HyClone, Logan, UT, USA) with 10% fetal bovine serum (FBS; HyClone), 1% penicillin/streptomycin (Gibco, Thermo Fisher Scientific, CA, USA) in a humidified 5% CO_2_ atmosphere at 37 °C. Activation was induced in PMA-differentiated THP-1 macrophages by stimulation with LPS, according to a previously described protocol [44]. Briefly, THP-1 cells (2 × 105 cells/mL in 100 μL) were seeded into a 96-well plate and differentiated into macrophage cells by stimulating with PMA (100 ng/mL) for 72 h, rest for overnight in fresh medium. The resulting macrophage cells were stimulated with 1 μg/mL LPS and incubated with or without diHEP-DPA or DHA at different concentrations for 48 h. Fifty µL of the sample was collected and centrifuged at 1000 rpm for 5 min, supernatants were tested for the secreted cytokines, IL-6, TNF-α, VEGF, and TGF-β1, using enzyme-linked immunosorbent assay (ELISA) kits (Abcam, Cambridge, UK).

### 2.3. Cancer Cell Culture and Collection of Conditioned Medium

The colorectal cancer cells, HT29 and HCT116 were grown in RPMI-1640 medium supplemented with 10% (*v/v*) FBS and 1% penicillin/streptomycin. Cells were plated at a density of 1 × 10^6^ cells in T75 culture flasks. Cancer cell-conditional medium (CCM) was generated by seeding HT29 (1 × 10^6^ cells/well) or HCT116 (5 × 10^5^ cells/well) cancer cell lines in 10 mm dishes and cultured in complete medium with 2% FBS for 24 h. Thereafter, the samples were collected and stored at −80 °C for further experiments.

### 2.4. TAM Polarization

THP-1 cells were differentiated into M0 (naïve) macrophages and TAMs [42]. THP-1 cells were treated with 100 ng/L PMA for 24 h and rest overnight. Following overnight resting, the cells were treated with CCM for 48 h, with or without diHEP-DPA. The CCM was collected daily (and replenished), then centrifuged and stored at −80 °C for further experiments. The collected media were termed tumor cancer conditioned medium (TCCM) or tumor cancer conditioned medium with added diHEP-DPA (TCCM-DPA). Images of differentiated THP1 cell morphology were acquired with an inverted light microscope.

### 2.5. Quantitative Measurement of Human Inflammatory Cytokines

Human inflammatory cytokines were measured using a BD Cytometric Bead Array human inflammatory cytokine assay kit (BD Biosciences, San Jose, CA, USA) as described by the manufacturer. Briefly, the same volume (50 µL) of mixed capture beads, sample, and PE detection reagent was added into a tube. The mixture was incubated for 3 h in dark. After washing and centrifuging, resuspended with 300 µL buffer, and analyzed using a fluorescence-activated cell sorting (FACS) system (BD Biosciences, San Jose, CA, USA).

### 2.6. Cell Proliferation

Proliferation rates of HCT116 and HT29 cells were measured using a commercial One Solution Assay Kit guided by the manufacturer’s instructions. Briefly, each cell line was seeded in a 96-well plate (1.5 × 10^4^ cells/well) and incubated with or without TCCM or TCCM-DPA for 24 h, after which 20 μL of a kit-provided solution was added. Plates were then incubated for 1 h, followed by measurement at OD490 using a microplate reader (Biotek, Seoul, Korea).

### 2.7. Phagocytosis Assay

Phagocytosis of human colorectal cancer cells (HT29 and HCT116) by THP-1 cells was assessed using a phagocytosis assay kit (Cayman Chemical, Ann Arbor, MI, USA). Cells were stained with carboxyfluorescein succinimidyl ester (CFSE) for 30 min. CytoBlue-labeled TAMs were then incubated overnight at 37 °C with CFSE-labeled colorectal cancer cells (HT29 and HCT116) at a 2:1 ratio. The FACS analysis was performed by first gating for CytoTel Blue-positive cells (i.e., macrophages), followed by gating for CFSE-positive macrophages. The FACS scattergram was generated by combining CytoBlue fluorescence with CFSE fluorescence; the double-positive population, corresponding to CFSE-labeled colorectal cancer cells phagocytosed by CytoBlue-labeled TAMs, is shown in the upper right quadrant.

### 2.8. Tumorspheres Formation

Single-cell suspensions of HT29 (5 × 10^5^ cells/well) and HCT116 (1 × 10^5^ cells/well) cancer cells were seeded in ultralow-attachment 6-well plates (Corning, Tewksbury, MA, USA) containing 2.5 mL of Cancer Stem Premium (Promab, Vancouver, BC, Canada) and incubated for 7 d. Tumorspheres were counted according to our previously described method [44]. The 6-well plate was scanned, and images were analyzed using NICE software. Tumorspheres formation was quantified by calculating tumorspheres formation efficiency (TFE, %). The effects of TCCM or TCCM-DPA on the TFE of colorectal cancer cells were assessed by pretreating HT29 and HCT116 cells with TCCM or TCCM-DPA for 2 d, after which cells were seeded in ultralow-attachment 6-well plates as described above; cells in the control group were incubated with complete culture medium. Direct effects of diHEP-DPA on the TFE of colorectal cancer cells were determined by adding 20 μM diHEP-DPA to the medium.

### 2.9. Scratch, Migration, and Invasion Assays

HT29 or HCT116 cancer cells (2 × 10^6^ cells) were inoculated into a 6-well plate. A scratch was made using a 200 μL pipette tip. Then the fresh RPMI-1640/0.5% FBS with 20 μM diHEP-DPA were added. After 24 h, images were acquired and percent inhibition was calculated relative to the control group. Invasion and migration assays were performed using a 24-well Transwell with 8 μm-pore polycarbonate membranes (Merck, Millipore, Darmstadt, Germany), with/without a Matrigel matrix basement coating (BD, San Jose, CA, USA) [45]. In the upper chamber, a suspension of HT29 (1 × 10^5^ cells/200 μL) or HCT116 (5 × 10^4^ cells/200 μL) cells treated with diHEP-DPA in RPMI-1640 with 0.5% FBS. In the bottom chamber, 900 μL RPMI-1640 with 20% FBS was added. After 24 h, fixed with 4% paraformaldehyde and stained with 0.03% crystal violet. Images were acquired using an inverted light microscope.

### 2.10. Flow Cytometric Analysis of ALDH Activity and CD47 Expression

After harvesting by treating with 1 × trypsin/EDTA and washing with 1 × PBS, cells (1 × 10^6^) were suspended and treated with diHEP-DPA (20 μM), TCCM, or TCCM-DPA for 48 h. Cells were incubated in 5 μL ALDH assay buffer at 37 °C for 30 min and assayed by FACS as described by the manufacturer. The negative control group was treated with the ALDH inhibitor diethylaminobenzaldehyde (DEAB). For CD47 expression determination, 5 μL of FITC-conjugated anti-human CD47 antibody (Life Technologies Corp, Carlsbad, CA, USA) was added and then incubated on ice for 30 min, followed by the addition of 300 μL of PBS for the assay by FACS.

### 2.11. Measurement of ROS Activity

Cancer cells were seeded into 96-well plates and treated with/without diHEP-DPA for 24 h. ROS were assayed using the redox-sensitive fluorescent dye, 2′,7′-dichlorofluorescein diacetate (DCFDA). After culturing diHEP-DPA-treated cancer cells, the cells were washed with 1 × PBS and incubated with 10 μM DCFDA for 30 min. Images were acquired by a phase-contrast fluorescence microscope (Lionheart FX Live Cell Imager; Biotek, Winooski, VT, USA), and analyzed using a FACS system.

### 2.12. Gene Expression Analysis

Total RNA was isolated using a TaKaRa MiniBEST Kit (TaKaRa, Tokyo, Japan) according to the protocol. Transcript levels were determined using a One-step AccuPower GreenStar RT-qPCR PreMix Kit with SYBR Green according to the manufacturer’s instructions (Bioneer Corporation, Daejeon, Korea). RT-PCR reactions were carried out in a total volume of 50 μL with 100 ng of RNA per reaction using specific primers described in Appendix A. The PCR cycling conditions are: 95 °C for 0.5 min, 55 °C for 0.5 min and 72 °C for 0.5 min, followed by a 10 min extension at 72 °C. A comparative CT method was used for the relative mRNA expression levels calculation. The β-actin gene was used as an internal control.

### 2.13. Western Blot Analysis

Proteins were isolated from tumorspheres or cancer cells, with and without 20 μM diHEP-DPA treatment, by lysing on ice for 45 min with lysis buffer containing a proteinase inhibitor cocktail, and followed by centrifugation at 12,000 rpm for 5 min. Cytosolic and nuclear proteins were isolated using a previously described method [46]. Isolated proteins were resolved on SDS-PAGE gels and transferred to a polyvinylidene difluoride (PVDF) membrane (Millipore, Bedford, MA, USA). PVDF membranes were blocked by incubating with 5% skim milk in Tris-buffered saline/Tween-20 (0.1%, *v/v*) (TBST) for 1 h, followed by incubation with primary antibodies at 4 °C overnight. All the antibodies as listed below were procured from Abcam and used for analysis. The list contained: anti-CD206 (ab64693), anti-CD163 (ab182422), anti-p65 (NF-κB) (ab16502), anti-phosphorylated p65 (p-p65; NF-κB) (ab76302), anti-SIRPα (ab191419), anti-CD47 (ab218810), anti-E-cadherin (ab40772), anti-N-cadherin (ab76011), anti-Vimentin (ab92547), anti-CD133 (ab216323), anti-CD44 (ab189524), anti-SOX2 (ab92494), anti-STAT3 (ab68153), anti-p-STAT3 (ab76315), anti-GAPDH (181602), and anti-laminB1 (ab16048). After washing, membranes were incubated with HRP-conjugated secondary antibodies (ab205718) and then developed with the ECL Plus Western blotting detection system (Pierce, Rockford) according to the manufacturer’s protocol. Immunoreactive proteins were detected by exposure of the PVDF blot to CL-XPosure film (Thermo Scientific, Rockford, IL, USA). Different exposure times were used to detect different proteins. Protein band density was determined by scanning densitometry and quantified using Image J (version 1.6, Maryland, MD, USA).

### 2.14. Detection of NF-κB (p65) and STAT3 Transcription Factor Activity

NF-κB (p65) and STAT3 DNA-binding activity in nuclear extracts from each group was detected using NF-κB (p65) and STAT3 Transcription Factor Assay Kits (Cayman Chemical). Detection procedures were strictly guided by the manufacturer. Briefly, 10 μL of nuclear extracts of samples was added to wells of a 96-well plate containing 90 μL of CTFB; for competition assays, 10 μL of nuclear extracts was added to 80 μL CTFB, followed by the addition of 10 μL of transcription factor NF-κB or STAT3 Competitor dsDNA per well. Plates were incubated overnight at 4 °C without shaking and washed five times with 1 × washing buffer. Thereafter, 100 μL of the primary antibody was added and plates were incubated at room temperature on an orbital shaker. Plates were then washed five times and transcription factor developing solution was added, followed by addition of stop solution was added and absorbance read at 450 nm.

### 2.15. Chemotherapy of Colorectal Cancer Cell-Bearing BALB/c Mice

Female BALB/c mice (6–8 week-old) were obtained from Orient Bio (Seongnam, Korea) and maintained in animal facilities for 1–2 weeks prior to experimentation. Total 40 mice were divided into four groups (*n* = 10/group): negative controls (Con), diHEP-DPA, 5-FU, and 5-FU+diHEP-DPA. Mice in the negative control group and diHEP-DPA group received no chemotherapy, and mice in the positive control group received 5-FU at an optimized dose of 20 mg/kg. Mice in experimental groups received an optimized diHEP-DPA dose of 20 μg/kg or 20 μg/kg diHEP-DPA combined with 20 mg/kg 5-FU. All drugs were administered by intraperitoneal injection. Body weight was measured three times per week. All animal experiments and procedures were conducted under a protocol approved by the Institutional Animal Care and Use Committee of the Korean Institute of Bioscience and Biotechnology (KRIBB-AEC-20005). After one month, the mice were sacrificed, and tumor tissues were obtained, photographed, and weighed. The dimensions of mouse tumors were measured and tumor volumes were calculated using the formula (width^2^ × length)/2 (Figure 6).

### 2.16. Statistical Analysis

All data are presented as the means ± standard deviation (SD). Data were analyzed by Student’s *t*-test using GraphPad Prism 8 Software (San Diego, CA, USA). A *p*-value < 0.05 was considered statistically significant.

## 3. Results

### 3.1. diHEP-DPA Suppresses Secretion of IL-6 and TNF-α by LPS-Stimulated Macrophage Cells

DiHEP-DPA is a novel resolvin that we previously synthesized using lipoxygenase derived from *Cyanobacteria* [43]. Figure 1 showed that diHEP-DPA inhibits LPS-induced IL-6 and TNF-α secretion in macrophages. The diHEP-DPA isolation method is summarized in Figure 1A. diHEP-DPA was generated from DHA with catalysis reaction of cyanobacterial lipoxygenase, which was identified, purified, and analyzed in our previous work. Then, the structure of diHEP-DPA was determined by HPLC, LC-MS/MS, and NMR. To elucidate the anti-inflammatory effects of diHEP-DPA, we stimulated PMA-differentiated THP1 macrophage cells with 1 μg/mL LPS to induce inflammation, then treated the cells with or without diHEP-DPA and DHA and measured secreted inflammatory cytokines by ELISA. As shown in Figure 1B, IL-6 and TNF-α secretion were significantly reduced in samples treated for 48 h with diHEP-DPA, which showed better anti-inflammatory effects than DHA at the same dose. Further analysis showed that this inhibition was diHEP-DPA concentration-dependent across a concentration range of 10–40 μM. These results show that diHEP-DPA has stronger anti-inflammatory effects than an equivalent concentration of DHA.

### 3.2. diHEP-DPA Modulates TAM Polarization via the NF-κB Signaling Pathway

Common methods for acquiring certain TAMs include isolating them from tumor tissues in vivo or perform a cancer cell-macrophage co-culture system in vitro [42]. In the current study, we determined whether diHEP-DPA modulates the polarization of TAM, results were shown in Figure 2. We treated naïve macrophages (M0) with a cancer cell-conditioned medium (CCM) for 48 h, with or without diHEP-DPA (20 μM) as shown in Figure 2A. After 48 h, CCM induced macrophage polarization, as evidenced by morphological changes; notably, diHEP-DPA significantly altered the polarization of TAMs (Figure 2B). We then assessed the expression of genes encoding CD163, CD206, and CD209—specific markers of TAMs (M2d type)—as well as expression of MMP2 and MMP9, triggering receptor expressed on myeloid cells 2 (TREM2) and vascular endothelial growth factor (VEGF). As shown in Figure 2C, CCM significantly increased expression of the TAM-related markers, CD206, CD163, VEGF, and TGF-β, at the protein level, whereas diHEP-DPA (20 μM) decreased their expression. We further investigated the secretion of the inflammatory cytokines, IL8, IL-1β, IL-6, IL-10, TNF-α, and IL12p70, by ELISA analysis of supernatants. This analysis showed that secreted IL-8 in CCM barely exceeded that in blank control beads, whereas supernatants of M0 macrophages contained higher amounts of secreted IL8, IL-1β, IL-6, and TNF-α, possibly reflecting the production of inflammatory cytokines by M0 macrophages, differentiated from THP1 cells by PMA treatment. M0 macrophages treated with CCM (TAMs) showed significantly increased secretion of IL-1β and IL-6 compared with untreated M0 macrophages, indicating that CCM successfully induced polarization of M0 macrophages into a TAM model. Interference of the polarization phase by diHEP-DPA clearly reduced the concentration of IL-1β, IL-6, and TNF-α. These results show that diHEP-DPA can modulate the polarization of TAMs, as evidenced by morphological changes, and changes at the gene-expression level, protein level, and even at the cytokine secretion level. To investigate the mechanism underlying this process, we probed the involvement of NF-κB, which plays a crucial role in immune responses, cellular growth, apoptosis, and inflammation; notably, NF-κB is also involved in regulating the transcription of cytokines, enzymes, and adhesion molecules associated with chronic inflammatory diseases [47,48]. As shown in Figure 2F, p65 protein levels in both cytosolic and nuclear fractions were significantly reduced in diHEP-DPA-treated TAMs compared with untreated TAMs, in association with an increase in the phosphorylated form of the protein (p-p65). Moreover, protein–nuclear binding assays performed to detect p65-DNA binding showed that diHEP-DPA reduced the ability of p65 protein to bind dsDNA (Figure 2G). The specificity of NF-κB–dsDNA binding was confirmed by performing assays in the presence of an excess of unlabeled self-competitor. These data suggest that diHEP-DPA inhibits the NF-κB signaling pathway during CCM-induced polarization of M0 macrophages into TAMs.

### 3.3. DiHEP-DPA Enhances TAM Phagocytic Activity by Blocking the CD47/SIRPα Axis

TAMs are an important tumor-promoting component of the TME that play a crucial role during malignant cancer progression and metastasis [49]. TAMs are found in stroma-rich primary colorectal cancer and facilitate proliferation, migration, invasion, EMT, and resistance; they also induce a CSC-like colorectal cell phenotype by reshaping the TME [50]. Thus, we investigated the effect of diHEP-DPA on the ability of TAMs to stimulate cancer proliferation, as shown in Figure 3. Briefly, we collected TCCM or TCCM-DPA, obtained during the TAM-polarization process, as described above. We then treated colorectal cancer cells with TCCM or TCCM-DPA for 2 d, then determined the cancer proliferation (Figure 3A). As shown in Figure 3B, diHEP-DPA treatment had no significant effect on the cell proliferation of HT29 or HCT116 colorectal cells. As expected, treatment with TCCM clearly enhanced the cell proliferation of colorectal cells, whereas this activity was significantly reduced by TCCM-DPA treatment compared with TCCM treatment alone.

TAMs are also phagocytic cells, which engulf and clear apoptotic cells, regulate the tumor immune micro-environment, and help the immune escape of cancer cells [51]. Following engulfment, TAMs increase the anti-inflammatory cytokines secretion and the production of Treg cells, at the same time, they inhibited the secretion of pro-inflammatory cytokines and enhanced the function of effector T cells [52]. As shown in Figure 3C, there was a notable enhancement of phagocytosis following diHEP-DPA treatment than the control group. As expected, the phagocytic ability of diHEP-DPA-treated TAMs was enhanced compared with that of untreated TAMs. Interestingly, abundant macrophage infiltration was found in CD47-positive tumor tissue. Upregulation of CD47, which interacts with SIRPα to prevent phagocytosis, is the mechanism that tumors enhance TAM polarization and escape macrophage-mediated damage [53]. Moreover, disruption of the CD47/SIRPα axis reduces the ability of tumor cells to escape phagocytosis. Accordingly, we evaluated the effect of diHEP-DPA on TAM expression of SIRP-α, colorectal cancer cell expression of CD47, and phagocytosis. As shown in Figure 3D, diHEP-DPA treatment significantly reduced SIRPα expression levels compared with that in untreated CCM-induced TAMs. Similarly, TCCM treatment up-regulated CD47 expression levels compared with that in the control group, possibly because of the higher concentration of inflammatory cytokines (IL-1β, IL-6, and TNF-α) in TCCM than in normal medium (Figure 2E). Moreover, diHEP-DPA treatment, as well as TCCM-DPA treatment, reduced the expression of CD47 in HT29 and HCT116 cells (Figure 3E,F). A previous mechanistic study reported that IL-6 derived from TAMs could increase the expression of CD47 in hepatoma cells via activation of the STAT3 pathway [54]. It has also been reported that inhibition of CD47 or SIRPα with antagonistic antibodies enhances the phagocytic activity of TAMs and suppresses tumor growth in preclinical models of glioblastoma [55], melanoma [56], lymphoma [57], and breast [58] and colorectal [59] cancer. These results establish diHEP-DPA as a potential anti-CD47 agent.

### 3.4. DiHEP-DPA Impedes TAM-Induced EMT and Acquisition of Stem-like Properties in CRCs

TAMs release a diverse array of growth factors, proteolytic enzymes, cytokines, and inflammatory mediators that are key agents in cancer stemness and metastasis, specifically contributing to tumor angiogenesis, growth, migration and invasion [60]. To evaluate the effects of diHEP-DPA on TAM-induced EMT and cancer stemness properties, CRCs were pretreated with TCCM or TCCM-DPA for 2 d, and analyzed for tumorspheres-formation assays (Figure 4). Figure 4A shows that TCCM treatment significantly increased both the number and size of tumorspheres compared with controls, an enhancement that was eliminated in the TCCM-DPA treatment group. These results are consistent with our demonstration that concentrations of IL-6, TNF-α, and IL-1β—all of which are vital factors for enhancing tumorigenesis, EMT, and cancer stemness of colorectal cells—were clearly higher in TCCM supernatants than in supernatants from the TCCM-DPA-treated group. We further investigated the effects of TCCM and TCCM-DPA on the migration and invasion of HT29 and HCT116 colorectal cancer cells. As shown in Figure 4B,C, TCCM significantly increased the number of cells that cross the wounded area in scratch-wound healing tests, results that were similar for Matrigel-coated (migration) and uncoated (invasion) Transwell membranes. We next investigated whether diHEP-DPA inhibited the EMT process. Specifically, we sought to determine whether diHEP-DPA inhibited the EMT process through modulation of TAM polarization by assessing expression of EMT-specific markers at both gene and protein levels. As illustrated in Figure 4D,E, expression of the epithelial marker E-cadherin was significantly reduced in TCCM-treated colorectal cancer cells, whereas expression of the mesenchymal markers N-cadherin and vimentin was increased, indicative of an enhanced EMT process. As expected, TCCM-DPA treatment reversed this enhancement.

EMT is widely associated with wound healing or tissue repair and angiogenesis, and recent studies have shown that cancer cells gain stem-like features after the EMT process [61]. CD133, CD44, and SOX2 are widely considered to be markers of colorectal cancer stem cells [62]. ALDH is known to be important in the protection of hemopoietic stem cells, and increased levels of ALDH activity were found in cancer stem cells [63]. Accordingly, we measured CD133, CD44, and SOX2 protein expression by Western blot and ALDH activity by FACS. This analysis showed that CD133, CD44, and SOX2 expression and ALDH activity were slightly increased in TCCM-treated colorectal cancer cells, whereas TCCM-DPA treated cancer cells exhibited a significant decrease in the expression/activity of these proteins (Figure 4F,G). These findings indicate that diHEP-DPA impedes TAM-induced acquisition of cancer stem-like properties by colorectal cancer cells. As noted above, there are substantial differences in the concentrations of IL-1β, IL-6, and TNF-α between TCCM and TCCM-DPA supernatants owing to diHEP-DPA modulation of the polarization of TAMs. Thus, both the EMT process and cancer stemness were reduced by TCCM-DPA treatment compared with TCCM treatment alone, whereas TCCM treatment mimics the true TME of colorectal cancer cells in vivo. These data indicate that the compound, diHEP-DPA, suppresses the EMT process and reduces cancer stem-like properties by modulating TAM polarization and the TME.

### 3.5. DiHEP-DPA Reduces Cancer Stemness by Increasing ROS Production

In addition to the effect of diHEP-DPA on cancer stemness through modulation of the polarization of TAMs, we were also interested in determining whether diHEP-DPA can interfere with the cancer stemness of colorectal cancer cells (Figure 5). To this end, we performed tumorspheres-formation tests and assessed the expression of CSC markers. As shown in Figure 5A, diHEP-DPA (20 µM) decreased tumorspheres numbers by 54% and reduced the size of the formed tumorspheres. Moreover, diHEP-DPA treatment reduced the protein expression of cancer stemness markers in colorectal cancer cells (Figure 5B). ROS levels tend to be lower in cancer stem cells than in differentiated cancer cells, and this less oxidative environment is actually needed to maintain the quiescence and self-renewal potential of cancer stem cells. It has thus been proposed that increased ROS could contribute to reducing the stemness and enhancing the differentiation of various stem cells. Accordingly, we measured ROS production in cancer stem cells treated with/without diHEP-DPA, and found that diHEP-DPA treatment increased the production of ROS without altering the cell viability (Figure 3B and Figure 5C,D). We hypothesized that diHEP-DPA-induced ROS could regulate the differentiation of cancer stem cells and reduce their stemness, drug resistance, and tumor angiogenesis. To address this possibility, we tested the effect of the ROS inhibitor, N-acetyl-L-cysteine (NAC), on tumorspheres formation in our system. Indeed, our results showed that NAC reversed the TCCM-DPA-induced reduction in the tumorspheres-formation efficiency of colorectal cancer cells (Figure 5E). ROS-mediated activation of the STAT3 signaling pathway was previously reported to be involved in cellular senescence [64]. To investigate the mechanism at the molecular level, we tested the STAT3 pathway in colorectal cancer cell-derived tumorspheres treated with 20 µM diHEP-DPA. As shown in Figure 5F,G, diHEP-DPA reduced levels of nuclear p-STAT3 compared to those in the control group, and NAC reversed the diHEP-DPA-induced dephosphorylation of STAT3.

### 3.6. DiHEP-DPA Inhibits Tumor Growth in a Xenograft Mouse Model

Given that diHEP-DPA showed modulating effects on TAM polarization in vitro, we investigated whether diHEP-DPA inhibited tumor growth in a subcutaneous tumor model in vivo (Figure 6). The bodyweights were not significantly changed by any treatments (Figure 6A), diHEP-DPA treatment alone decreased tumor volume and weight compared with that in the control group (Figure 6B,C). Moreover, tumor volumes and weights in the group treated with diHEP-DPA and 5-FU (5-FU+diHEP-DPA group) were lower than those in the 5-FU-treated group. We further found that diHEP-DPA effectively reduced concentrations of the inflammatory cytokines IL-6 and TNF-α in serum (Figure 6D), alone or in combination with chemotherapy. These results demonstrate that diHEP-DPA effectively inhibits tumorigenicity in a subcutaneous colorectal cancer model, and is more effective in combination with chemotherapy than chemotherapy alone.

## 4. Discussion

Colorectal cancer is often diagnosed at an advanced stage and chemotherapy or targeted therapies improved overall survival for patients to some extent. The poor clinical outcomes find their origin primarily in therapeutic resistance, with immunosuppression mechanisms to both chemotherapy and targeted therapy remaining the key culprits [65]. Colorectal cancer cell development and progression are complex processes caused by the TME, which recruits vasculature and stroma, including immune cells, fibroblasts, cytokines, and the extracellular matrix that surrounds them [4].

Specialized pro-resolving mediators and their precursors have been shown to exert antitumor activities against various cancers through multiple mechanisms and targets, including angiogenesis, EMT, pro-tumorigenic cytokines, cancer stem cells, natural killer cells, and macrophages. Many studies have also reported that resolvins induce a switch in the polarization of macrophages from M1 to M2 [66,67,68], and inhibit TAM polarization [42]. Our group has attempted to develop novel and economic resolvins with anti-inflammatory actions. In our previous work, we synthesized the novel anti-inflammatory 7S,15R-dihydroxy-16S,17S-epoxy-docosapentaenoic acid (diHEP-DPA). In the current study, we verified the anti-inflammatory activity of diHEP-DPA, showing that it reduces the secretion of IL-6 and TNF-α in LPS-induced macrophage inflammation (Figure 1). Because an inflammatory TME is crucial for the polarization of TAMs in colorectal cancer, we were interested in determining whether diHEP-DPA has the potential to regulate the polarization of TAMs. We found that diHEP-DPA reduced the expression of TAM markers (CD206, CD163, and VEGF), decreased the secretion of IL-1β, IL-6, and TNF-α in supernatants, and modulated polarization during CCM-induced differentiation (Figure 2). Activation of NF-κB produces proinflammatory cytokines, providing an important link between inflammation and cancer [69,70] that has been effectively demonstrated in both colon and liver cancer [71]. Accordingly, we speculated that diHEP-DPA inhibited p65 during modulation of TAM polarization, a supposition confirmed by evidence that diHEP-DPA reduced expression of p65 protein in both cytosolic and nuclear compartments levels of p-p65(incorrect), thus reducing p65 localization and binding to target DNA sequence. These results are consistent with previous reports showing that polarization of macrophages to TAMs requires NF-κB activation.

TAMs interact with tumor cells and make major contributions to tumor progression and resistance, reflecting their important role in the TME [72]. To investigate the effects of TAM polarization induced by CCM, with/without diHEP-DPA, on the various properties of colorectal cancer cells, we treated HT29 and HCT116 cells with TCCM or TCCM-DPA, or directly treated cancer cells with diHEP-DPA. As expected, TCCM treatment significantly increased the proliferation of colorectal cells. Interestingly, TCCM-DPA effectively reversed this process (Figure 3B). These results are well explained by the idea that diHEP-DPA might dampen the oncogenic crosstalk between cancer cells and macrophages, interfering with the secretion of inflammatory cytokines that accompanies macrophage polarization to TAMs and promoting the proliferation of cancer cells.

TAMs can affect cancer progression is through phagocytosis of tumor cells. Phagocytes rapidly recognize and engulf apoptotic cells before their release of intracellular components. This maintains the membrane integrity of apoptotic cells, preventing exposure to potentially immunogenic materials and subsequent inflammatory responses [51]. This physiological process efficiently removes apoptotic cells without subsequent secondary necrosis or damage [73]. TAMs are a type of phagocyte, and as such are involved in phagocytosis [74,75]. Our results demonstrated that diHEP-DPA enhanced phagocytosis by modulating the polarization of TAMs (Figure 3C). Another mediator of tumor immune evasion in addition to TAMs is the CD47/SIRPα axis. SIRPα contains tyrosine-based motifs, SHP-1 (Src homology 2 [SH2] domain-containing protein tyrosine phosphatase) and SHP-2. In the absence of CD47 binding to SIRPα, the resulting failure to recruit SHP-1 and SHP-2 enables activation of phagocytosis [76]. This is illustrated in Figure 3D–F, which shows that diHEP-DPA blocked the CD47/SIRPα axis by directly or indirectly reducing the expression of these proteins, and thus enhanced the phagocytosis of live, intact colorectal cancer cells (HT29 and HCT116) by TAMs.

Additionally, TAM-secreted proinflammatory cytokines, such as IL-6 and TNFα, contribute to “dormant inflammation”, which determines immunosuppression in the TME. Macrophages secrete various soluble cytokines and inflammatory mediators that are not only involved in tumor angiogenesis, but also promote the EMT process and cancer stemness [61]. We verified this, showing that diHEP-DPA eliminated the promotive effect of TCCM on colorectal tumorspheres formation, migration, and invasion (Figure 4A–C). It has been reported that EMT remarkably enhances the metastatic potential and invasion of cancer cells, and the resulting mesenchymal-like cancer cells are resistant to cancer therapy [77]. Thus, we further assessed EMT status after treatment with TCCM or TCCM-DPA, demonstrating that TCCM enhanced the EMT process by increasing the expression of N-cadherin and vimentin and decreasing the expression of E-cadherin (Figure 4D and E). After going through the EMT process, cancer cells express high levels of stem-like cell markers [78,79]. Our results, shown in Figure 4F,G, also verified that TCCM enhanced the EMT process and helped cancer cells gain stem-like properties, as evidenced by the expression of cancer stemness markers (CD133, CD44, SOX2, and ALDH), whereas diHEP-DPA reversed this effect.

In addition to the indirect effects of diHEP-DPA on colorectal cancer cells through modulation of TAMs, we were also curious about whether diHEP-DPA regulates the EMT process and/or cancer stemness directly. As shown in Figure 5, treatment with diHEP-DPA (20 μM) reduced tumorspheres-formation efficiency by ~40%, and significantly reduced specific markers of cancer stemness. These results are consistent with our previous study, which showed that diHEP-DPA inhibited breast cancer stemness [42]. CSCs maintain lower levels of ROS than differentiated cancer cells, which is actually needed to keep quiescence and the self-renewal ability of CSCs [80,81]. Similarly, diHEP-DPA increased the production of ROS in colorectal cancer cells without altering their viability, which argues against the concept that increasing ROS in cancer cells should kill these cells. However, there is no evident dose–response relationship between cellular ROS level and cytotoxicity. Our observation that diHEP-DPA increased ROS production is compatible with previous findings that lactic acidosis and L-buthionine sulfoximine induce increases in ROS levels without negatively impacting the growth of various cancer cell lines. Moreover, increased ROS could contribute to reducing the stemness and enhancing the differentiation of stem cells, as suggested by previous reports from Sato’s research group that H_2_O_2_ induces differentiation of glioblastoma stem cells by increasing intracellular ROS without substantially reducing viability [82]. Zhao et al. reported that the differentiation of liver cancer stem cells was promoted when the cells were exposed to exogenous ROS [83]. Moreover, one of the cellular senescence mechanisms is that ROS mediated the activation of the STAT3 [64]. Our results illustrated that diHEP-DPA reduced the protein levels of nuclear p-STAT3 and STAT3 DNA-binding activity (Figure 5F,G), suggesting that diHEP-DPA inhibited the formation of tumorspheres by generating ROS and inhibiting the STAT3 signaling pathway.

## 5. Conclusions

DiHEP-DPA is a novel specialized pro-resolving mediator synthesized previously by our group. In this study, we explored its potential function in TME-targeted therapy, including its effects on TAMs, immunosuppression, the EMT process, and cancer stemness. Our results showed that diHEP-DPA effectively reduced tumor volume in vivo and inhibited immunosuppression, the EMT process, and cancer stemness by modulating the polarization of TAMs in vitro. Mechanistically, diHEP-DPA reduced the expression of TAM markers by inhibiting the NF-κB signaling pathway, and inhibited cancer stemness via the ROS/STAT3 signaling pathway. Furthermore, diHEP-DPA blocked the CD47/SIRPα axis, thereby enhancing phagocytosis. Collectively, these results demonstrate that diHEP-DPA could be developed as an anticancer agent against colorectal cancer.

## Figures and Tables

**Figure 1 antioxidants-10-01459-f001:**
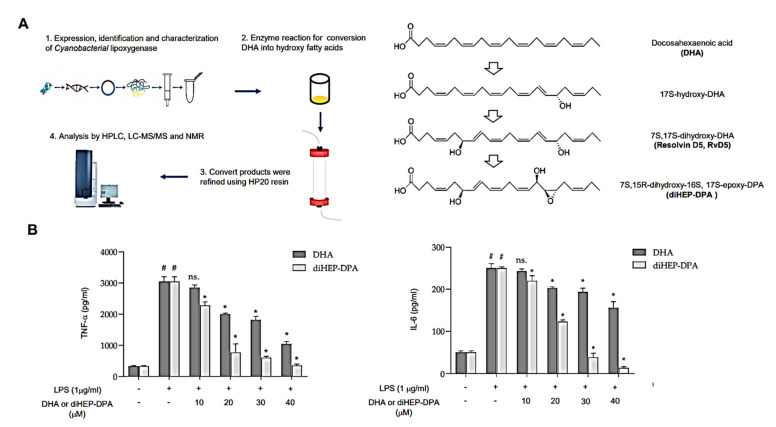
DiHEP-DPA inhibits LPS-induced IL-6 and TNF-α secretion in macrophages. (**A**) Summary of the synthesis process of diHEP-DPA. (**B**) Production of IL-6 and TNF-α by LPS-stimulated (inflamed) macrophages was inhibited by different concentrations of diHEP-DPA and DHA. Cytokines were measured by ELISA. Data from triplicate experiments are presented as the means ± SD (^#^
*p* < 0.05 versus the DMSO-treated control group; * *p* < 0.05 versus the LPS-treated positive control group).

**Figure 2 antioxidants-10-01459-f002:**
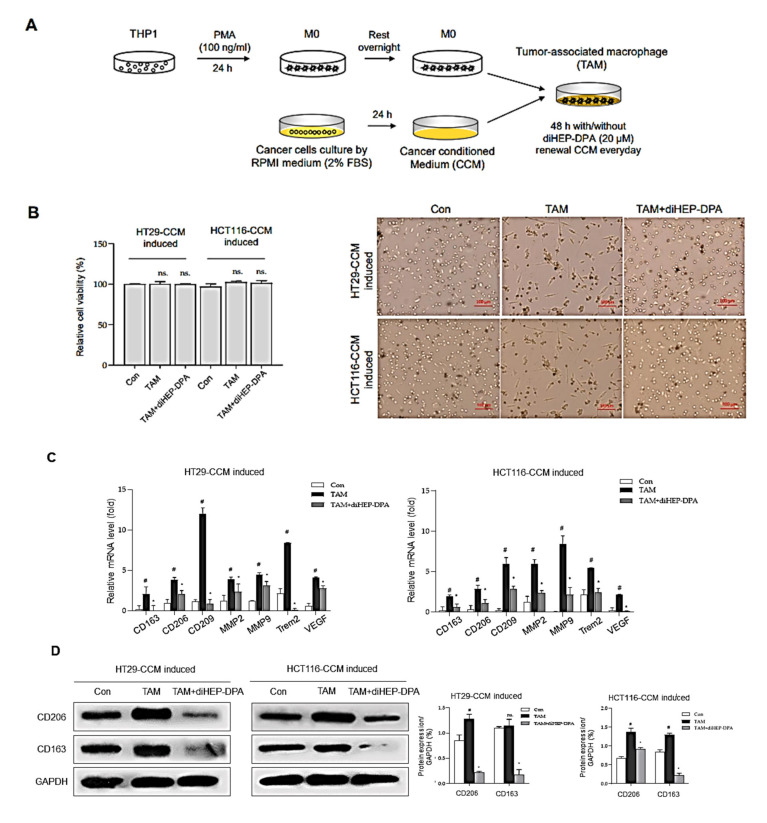
DiHEP-DPA modulates TAM polarization. (**A**) Experimental procedure for inducing TAMs. (**B**) diHEP-DPA–induced morphological changes after 48 h without damage to cell viability. Scale bar: 100 μm. (**C**) Transcript levels of the TAM markers, CD133, CD206, CD209, MMP2, MMP9, TREM2, and VEGF were determined in diHEP-DPA-treated macrophage using gene-specific primers and quantitative RT-PCR. β-actin was detected as an internal control. (**D**) CD206 and CD163 expression, assessed by Western blotting using antibodies against CD206 and CD163. DiHEP-DPA decreased CD206 and CD163 expression during TAM polarization. (**E**) Cytokine profile assay of conditioned media from TAMs and diHEP-DPA-treated TAMs, performed using a specific ELISA kit for VEGF and TGF-β1, and cytokine beads. (**F**) NF-κB activation was assessed in TAMs treated with/without diHEP-DPA using antibodies against p65, p-p65, GAPDH, and lamin B1. DiHEP-DPA decreased protein expression of p65 in both nuclear and cytosolic compartments during TAM polarization. (G) Nuclear protein binding assay showing binding of nuclear NF-κB with DNA in lysates from TAMs treated with diHEP-DPA, determined using an NF-κB Transcription Factor Assay Kit (Cayman). Data from triplicate experiments are presented as the means ± SD (*^#^ p* < 0.05 versus the DMSO-treated control group; ** p* < 0.05 versus the TAM group).

**Figure 3 antioxidants-10-01459-f003:**
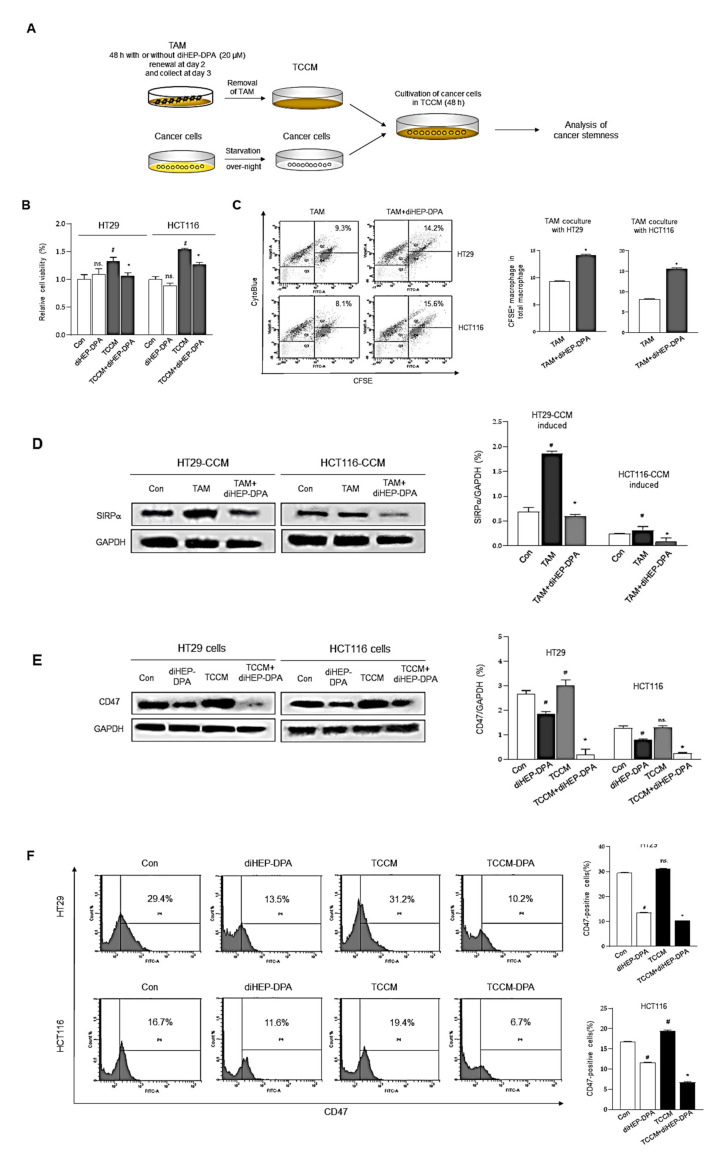
DiHEP-DPA effects on the proliferation and immunosuppression of colorectal cancer cells through modulation of TAM polarization. (**A**) Experimental procedure for treatment of colorectal cells with conditioned medium from TAMs treated without (TCCM) or with (TCCM-DPA) diHEP-DPA. (**B**) DiHEP-DPA did not significantly inhibit the proliferation of HT29 or HCT116 cells. (**C**) Restoration of TAM phagocytic activity towards apoptotic HT29 and HCT116 cells by diHEP-DPA. CytoBlue-labeled TAMs were incubated with CFSE-labeled HT29 and HCT116 colorectal cancer cells and analyzed by FACS. (**D**,**E**) SIRPα expression in TAMs and CD47 expression in colorectal cancer cells, determined by Western blotting using antibodies against SIRPα and CD47. (**F**) Decrease in the CD47-positive cell population by TCCM-DPA treatment. Colorectal cancer cells were treated with TCCM or TCCM-DPA for 48 h and subjected to FACS analysis. Data from triplicate experiments are presented as the means ± SD (^#^
*p* < 0.05 versus the DMSO-treated control group; * *p* < 0.05 versus the TCCM group).

**Figure 4 antioxidants-10-01459-f004:**
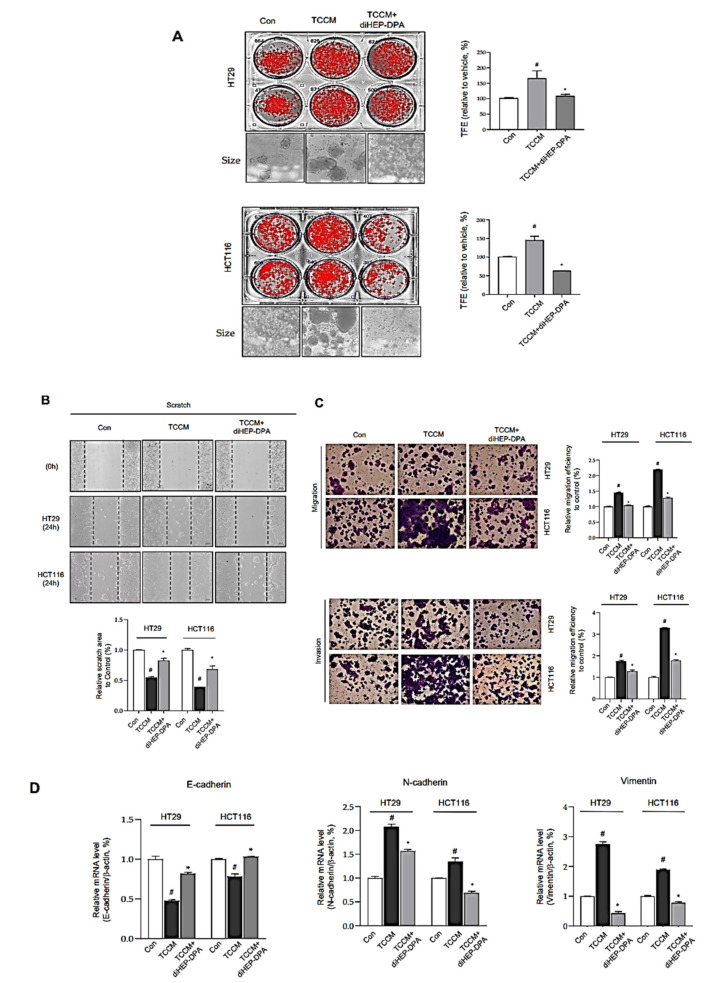
DiHEP-DPA affects the EMT process and stemness of colorectal cancer cells through modulation of TAM pola ization. (**A**) Decreased tumorspheres-formation efficiency (TFE) by TCCM-DPA treatment. Tumorspheres derived from HT29 and HCT116 cells were cultured for 7 d in the presence of TCCM, TCCM-DPA, or DMSO. (**B**) Migration of HT29 and HCT116 cells treated with TCCM or TCCM-DPA (RPMI-1640/0.5% FBS), determined by scratch assay. Scale bar: 100 μm. (**C**) Cell migration (without Matrigel) and invasion (with Matrigel) of HT29 and HCT116 cells exposed to TCCM or TCCM-DPA, determined by Transwell assays. Scale bar: 100 μm. (**D**) Transcript levels of the EMT markers, E-cadherin, N-cadherin, Vimentin in TCCM-treated HT29, and HCT116 cells, determined using gene-specific primers and quantitative RT-PCR. β-actin was detected as an internal reference. (**E**,**F**) Expression of EMT protein markers and cancer stemness markers (CD133, CD44, and Sox2), determined by Western blotting using the corresponding antibodies. (**G**) Decrease in the ALDH-positive cell population by treatment with TCCM-DPA. Colorectal cancer cells were treated with TCCM or TCCM-DPA for 48 h and analyzed by FACS. Data from triplicate experiments are presented as the means ± SD (^#^
*p* < 0.05 versus the DMSO-treated control group; * *p* < 0.05 versus the TCCM group).

**Figure 5 antioxidants-10-01459-f005:**
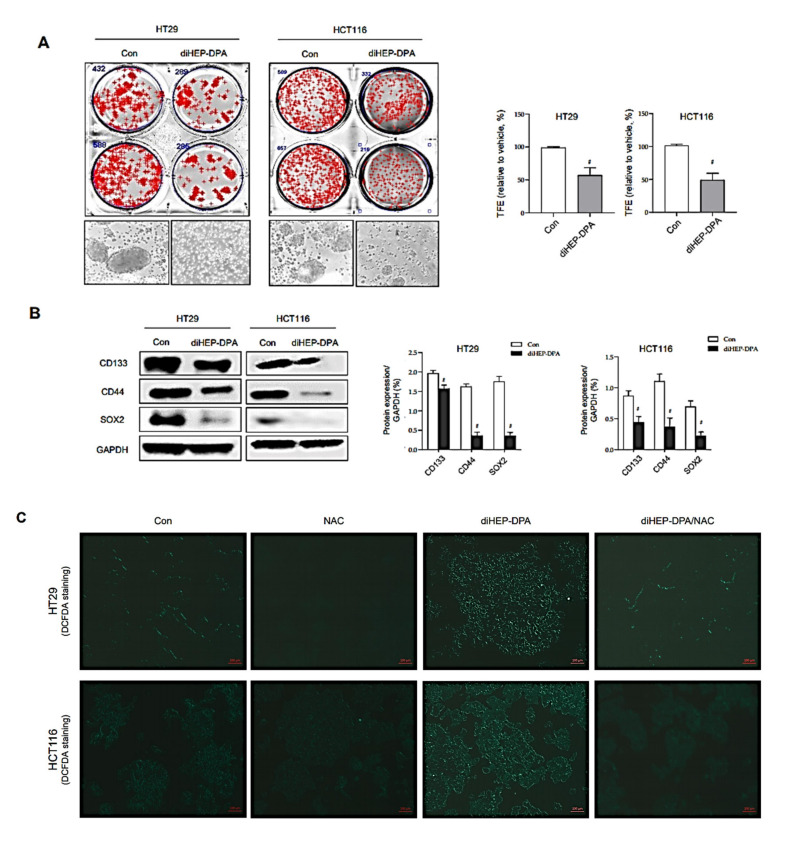
DiHEP-DPA directly affects the stemness of colorectal cancer cells through the ROS/STAT3 signaling pathway. (**A**) Decrease in tumorspheres-formation efficiency (TFE) by treatment with diHEP-DPA (20 μM). Tumorspheres derived from HT29 and HCT116 cells were cultured for 7 d in the presence of diHEP-DPA or DMSO. Microscopy image shows the sizes of representative tumorspheres. Scale bar: 100 μm. (**B**) Expression of cancer stemness markers (CD133, CD44, and Sox2), determined by Western blotting using the corresponding antibodies. DiHEP-DPA reduced the expression of cancer stemness markers. (**C**,**D**) Effect of diHEP-DPA (20 μM) on ROS generation in HT29 cells, determined using DCF-DA staining. Scale bar: 100 μm. (**E**) Representative images obtained under 10× magnification showing reversal of TCCM-DPA–induced reduction in tumorspheres-formation efficiency by the antioxidant, NAC. Tumorspheres formation was determined after 7 d. Scale bar: 100 μm. (**F**) Effects of diHEP-DPA and NAC on STAT3 phosphorylation. STAT3 activation in tumorspheres, determined by Western blotting using antibodies against pSTAT3, STAT3, GAPDH, and lamin B. DiHEP-DPA decreased the nuclear levels of p-STAT3 in tumorspheres, and the diHEP-DPA-induced dephosphorylation of p-STAT3 was ameliorated by NAC. (**G**) Nuclear protein binding assay showing binding of p-STAT3 with DNA in lysates from tumorspheres treated with diHEP-DPA, determined using a STAT3 Transcription Factor Assay Kit (Cayman). DiHEP-DPA reduced the binding of nuclear p-STAT3 with DNA. Data from triplicate experiments are presented as the means ± SD (^#^
*p* < 0.05 versus the DMSO-treated control group; ** p* < 0.05 versus the TCCM group.

**Figure 6 antioxidants-10-01459-f006:**
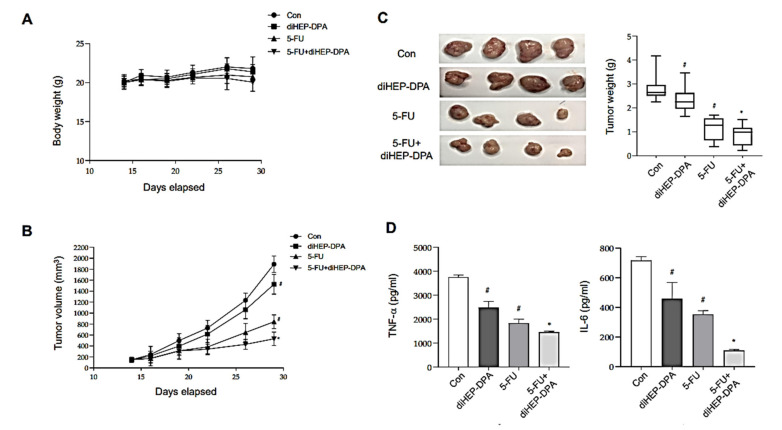
Effect of diHEP-DPA on tumor growth in a xenograft model. A CT26 subcutaneous colorectal tumor model was established by injecting 5 × 10^6^ cells into a female BALB/c mouse. Mice were divided into four groups: con (saline), diHEP-DPA (20 μg/kg), 5-FU (20 mg/kg), and 5-FU + diHEP-DPA. (**A**) Changes in the body weight of tumor-bearing mice with different therapies. (**B**,**C**) Tumor volume and tumor weights in mice with different therapies. After 2 weeks, mice were sacrificed and tumor specimens were collected, photographed, and weighed. Tumor dimensions were measured two or three times per week using a caliper, and volume was calculated as width^2^ × length/2. Tumor growth was monitored throughout the experimental period. Tumor weights were measured after therapy. (**D**) Concentrations of inflammatory cytokines (TNF-α and IL-6) in serum, assayed on day 30. Data are presented as the means ± SD of three independent experiments (*^#^ p* < 0.05 versus the DMSO-treated control group; ** p* < 0.05 versus the 5-FU group).

## Data Availability

Data is contained within the article and Appendix A.

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
