# Peer review of "7S,15R-Dihydroxy-16S,17S-Epoxy-Docosapentaenoic Acid, a Novel DHA Epoxy Derivative, Inhibits Colorectal Cancer Stemness through Repolarization of Tumor-Associated Macrophage Functions and the ROS/STAT3 Signaling Pathway"

_antioxidants, 2021, doi:10.3390/antiox10091459_

Round 1

Reviewer 1 Report

Introduction: the introduction provides a very good and concise overview of the main topics discussed in the manuscript, including the function of M1 and M2 macrophages, EMT, and cancer stem cells, also including the resolvins to prepare the reader for the experimental part. Also, the methods are all described in a precise and concise fashion, leaving no major questions about the protocols and their relevance. Overall, the spectrum of different methods used in this manuscript is amazing, ranging from immunology- and multiple cell line-based experiments to animal studies. 

The efficacy of the novel resolvin on cytokine secretion is amazing (compared to DHA), what would be an assumed EC50 value in comparison to DHA? And the structure of the molecule containing an active epoxy ring raises the question about the chemical stability of the molecule: what's the half-life of this molecule in aqueous solutions (and the cellular cytoplasm)? Are the active concentrations of 10 - 40 uM achieved (and for how long) in cell culture conditions, what's the solubility, and how fast (and by which mechanisms) is it cleared by cells? 

Figure 2: the morphologic changes in macrophages (formed THP1 cells)  induced by the conditioned medium is not really visible in Fig. 2B, nor is their inhibition by the substance. I would be highly recommended to use higher-resolution, and higher magnification images. The remainder of the elements in F2 is nevertheless convincing, especially the changes in biomarker expression levels. It's also good that mRNA and protein expression was probed for the most relevant molecules. The same applies to the detection of the various secreted cytokines and chemokines. Last not least, Western blots are largely of high quality; I do not know if the journal requires submission of the complete blots as reference and control of significance, but it is very positive to see that these are provided indeed ("non-published data"). The quality of the complete blots also looks rather convincing. 

It gets more complex with Fig. 3, but the single elements are all rather convincing, even if some of the effects observed (such as in Fig. 3B) are not very pronounced. The question is if these 2 colorectal cancer cell lines are an optimal choice for the immunologic activation investigated here. Have other cell lines been explored? Are these the CRC cell lines that show the highest response, are they characteristic for other cell lines? Please describe a bit more in detail why these cell lines were chosen, on which basis. Relative quantification of the SIRPa expression by Western blotting and GAPDH ratio looks very convincing and significant. 

Figure 4: some of the elements of Fig. 4 could also benefit from higher resolution, or at least higher contras (especially Fig. 4B). In Fig. 4D, it would be highly beneficial if the name of the investigated genes were written more prominently (and in larger letters) above or below the actual plots. The data from 4G however makes you wonder if there is a significant number of stem cells detectable under these specific cell culture conditions (2D monolayer) and if these are the right conditions to investigate the issue. 

Fig. 5: I am not convinced of the relevance and quality of the tumorspheres assay. The images provided (Fig 5A and B, 5E) are not suitable to get the basic phenotype of these "spheres" across, it rather looks like a colony-formation assay. Please provide better images for this assay. Even more problematic is Fig 5C: I cannot see anything in these images, and certainly no specific DCFDA staining. This has to be replaced by higher-quality images and cannot be published in this form. I would anyway question the relevance of these tumorsphere experiments: they essentially result in partial de-differentiation of the colorectal cancer cells, and may increase the degree of stemness and associated EMT; although the functional relevance of this may not be fully physiological and therefore potentially misleading. It would be better to use matrix-embedded 3D cultures to assess these questions, as under these conditions more physiologically relevant phenotypes are achieved. 

Fig. 6: the mouse xenografting data are rather dramatic and convincing, there is nothing to add to these. 

Overall, I think that the effects of the compound on tumor-immune cell interactions (and on tumor cells alone) are quite convincingly demonstrated, although I may not 100% agree with some of the conclusions concerning the molecular and cellular mechanisms. I am not convinced that the used methods are fully appropriate to investigate the role of EMT, as additional experimental settings may be required to demonstrate this. It is also debated what role EMT plays in colorectal cancer progressions, but even more, if the chosen cell lines are a good model to investigate EMT? It is good that the discussion is mainly focused on the inflammatory aspects which are also demonstrated convincingly by the experiments performed here. The same applies to the ROS aspects. If the elements of EMT and stemness are so critical for this discussion, is another question (as outlined above). 

minor things: 

line 86: the sentence is incomplete: "In many colorectal cancer patients, can initially achieve a good therapeutic effect." 

line 417-418: "TAMs are founded in stroma-rich primary colorectal cancer and facilitate proliferation, migration, invasion, epithelial-mesenchymal transition (EMT) and therapeutic resistance" - it probably should mean "found" 

Author Response

Dear Reviewer:

Thank you for your letter and comments concerning our manuscript entitled “7S,15R-dihydroxy-16S,17S-epoxy-Docosapentaenoic Acid, a Novel DHA Epoxy Derivative, Inhibits Colorectal Cancer Stemness through Repolarization of Tumor-Associated Macrophage Functions and the ROS/STAT3 Signaling Pathway” (Antioxidants-1372403).

We greatly appreciate your positive comments. Those comments are all valuable and very helpful for revising and improving our paper, as well as the important guiding significance to our researches. We have studied comments carefully and have made correction which we hope meet with approval. Revised portion are marked in red in the paper. The main corrections in the paper and the responds to your comments are as attached files.

Reviewer 2 Report

The manuscript entitled “7S,15R-dihydroxy-16S,17S-epoxy Docosapentaenoic Acid, a Novel DHA Epoxy Derivative, Inhibits Colorectal Cancer Stem-ness through Repolarization of Tumor-Associated Macrophage Functions and the ROS/STAT3 Signaling Pathway” by Wang L. et al. described the anti-inflammatory role of diHEP-DPA in colorectal cancer, focusing on tumor associate macrophages polarization and colorectal cancer cell stemness. The topic is of interest and should be considered for publication in Antioxidants Journal, but some points must be addressed before publication.

Minor points:

- Page 2 line 51: “...some problems remain” please correct this sentence in a more scientific language.

- Page 3 lines 129-132 are very similar to previous published paper of the authors (https://doi.org/10.3390/antiox10030457). Please re-write this part.  

- Page 4 line 162: please specify the percentage of serum used.

- Page 4 line 164: Instead of inflammation I suggest writing “activation” of THP1 cells.

- Figure 2B is of poor quality and too small. Please provide images of better quality.

- Does DiHEP-DPA affect cell viability of TAM? This data should be added.

- Figure 4A and 5A are of poor quality and differences in tumor spheres size are not detectable. If this data is important for authors, a graph reporting size differences should be added.

- Figure 4B. The authors should quantify the rate of recovery of the scratch by reporting the data of the scratch area measured.

- Figure 4C. The authors should quantify the invasion/migration of cancer cells (i.e. measuring absorbance of eluted crystal violet).

- Figure 4F. Please correct CD144 in the figure with CD44.

- Figure 5C. Fluorescence images are of poor quality. Please provide images of better quality.

- Page 16 line 552. I suggest indicating in the text also Figure 3B that reports data about cell proliferation after DiHEP-DPA treatment.

- Figure 5G caption (page 18 line 582). STAT3 binding assay regards colorectal cancer cell and not TAM, isn’t it?

- Figure 6A and B. Curves labels are not present.

- Figure 6D caption (page 18 line 607). Please correct, in this figure group TCCM is not reported.

- In in vivo experiments the authors should evaluate if there are difference between groups in macrophages polarization and amount. Does DiHEP-DPA treatment impact on macrophage polarization also in vivo?

- Page 21 line 731-733. These sentences are a typo of the “template”. Please remove it.

- This manuscript would be improved by an overall revision of English.

Author Response

(The authors gave the same response as above.)
